# REM-Predominant Obstructive Sleep Apnea in Patients with Coronary Artery Disease

**DOI:** 10.3390/jcm11154402

**Published:** 2022-07-28

**Authors:** Baran Balcan, Yeliz Celik, Jennifer Newitt, Patrick J. Strollo, Yüksel Peker

**Affiliations:** 1Department of Pulmonary Medicine, Koç University Hospital Istanbul, 34010 Zeytinburnu, Turkey; 2Research Center for Translational Medicine, Koç University School of Medicine, 34450 Istanbul, Turkey; yecelik@ku.edu.tr; 3Division of Pulmonary, Allergy, and Critical Care Medicine, University of Pittsburgh School of Medicine, Pittsburgh, PA 15213, USA; newittjl@upmc.edu (J.N.); strollopj@upmc.edu (P.J.S.J.); 4Division of Sleep and Circadian Disorders, Brigham and Women’s Hospital, Harvard Medical School, Boston, MA 02115, USA; 5Department of Molecular and Clinical Medicine, Institute of Medicine, Sahlgrenska Academy, University of Gothenburg, 405 30 Gothenburg, Sweden; 6Department of Clinical Sciences, Respiratory Medicine and Allergology, Faculty of Medicine, Lund University, 221 84 Lund, Sweden

**Keywords:** obstructive sleep apnea, coronary artery disease, REM sleep, depression, quality of life

## Abstract

Obstructive sleep apnea (OSA) is common in adults with coronary artery disease (CAD). OSA that occurs predominantly during rapid-eye movement (REM) sleep has been identified as a specific phenotype (REM-predominant OSA) in sleep clinic cohorts. We aimed to examine the association of REM-predominant OSA with excessive sleepiness, functional outcomes, mood, and quality of life in a CAD cohort, of whom 286 OSA patients with total sleep time ≥ 240 min, and REM sleep ≥ 30 min, were included. REM-predominant OSA was defined as a REM-apnea-hypopnea-index (AHI) /non-REM (NREM) AHI ≥ 2. In all, 73 (25.5%) had REM-predominant OSA. They were more likely to be female (26.0% vs. 9.9%; *p* = 0.001), and more obese (42.5% vs. 24.4%; *p* = 0.003) but had less severe OSA in terms of AHI (median 22.6/h vs. 36.6/h; *p* < 0.001) compared to the patients with non-stage specific OSA. In adjusted logistic regression models, female sex (odds ratio [OR] 4.64, 95% confidence interval [CI] 1.85–11.64), body-mass-index (BMI; OR 1.17; 95% CI 1.07–1.28) and AHI (OR 0.93, 95% CI 0.91–0.95) were associated with REM-predominant OSA. In univariate linear regression models, there was a dose-response relationship between REM-AHI and Zung Self-rated Depression Scale but not excessive sleepiness, functional outcomes, and anxiety scores. Among the Short Form-36 subdomains, Vitality, Mental Health, and Mental Component Summary (MCS) scores were inversely correlated with REM-AHI. In multivariate linear models, only MCS remained significantly associated with REM-AHI after adjustment for age, BMI, and sex (β-coefficient −2.20, %95 CI [−0.56, −0.03]; *p* = 0.028). To conclude, female sex and BMI were related to REM-predominant OSA in this revascularized cohort. MCS was inversely associated with REM-AHI in the multivariate model.

## 1. Introduction

Obstructive sleep apnea (OSA) is a chronic condition characterized by repetitive episodes of cessation of airflow and arousals from sleep and intermittent hypoxemia [1]. Exhibition of OSA differs significantly between individuals in clinical cohorts; i.e., not all OSA patients demonstrate excessive daytime sleepiness (EDS) and increased risk of cardiovascular morbidity [2]. Further exploration of these differences is therefore crucial for a better understanding of the features, prognosis, and cardiovascular outcomes in patients with OSA. 

Rapid eye movement (REM) sleep, which typically accounts for 20–25% of total sleep time, is associated with distinct physiological variations that influence upper airway function [3]. During REM sleep, there is a tendency for upper airway collapse due to cholinergic-mediated suppression of genioglossus activity [4,5]. REM sleep is also associated with increased hemodynamic variability, sympathetic activity, and myocardial demand [6]. OSA that occurs predominantly during REM sleep has been identified as a specific phenotype (REM-predominant OSA) and has been reported to occur among 14% to 36% of all OSA cases [7]. OSA events during REM sleep are typically longer, more frequent, and are related to greater oxyhemoglobin desaturation than events during non-REM sleep [6]. Patients with REM-predominant OSA tend to be younger women and have a less severe OSA in terms of the apnea-hypopnea index (AHI) [3]. Changes in sleep architecture regarding REM sleep have been associated with depressive mood [8,9] and anxiety [10], but not with impairment in quality of life [11]. REM-predominant OSA has been independently related to prevalent and incident hypertension as well as impaired glucose metabolism [12,13]. A dose-response relationship between REM-AHI and non-dipping blood pressure has also been reported [14]. 

Coronary artery disease (CAD) is one of the most common conditions associated with increased morbidity and mortality [10]. The occurrence of OSA among patients with CAD is very high (50% as compared to 10–20% in the general adult population) [2], and OSA patients have an increased risk for incident CAD compared to individuals without OSA [2,15]. Moreover, major adverse cardiovascular and cerebrovascular events have been reported to be more common in patients with CAD with concomitant OSA [16]. Less is known regarding the determinants of REM predominant OSA and its associations with EDS, functional outcomes, anxiety, depression, and quality of life in adults with CAD. 

The “Randomized Intervention with CPAP in Coronary Artery Disease and Obstructive Sleep Apnea” (RICCADSA) trial primarily addressed the impact of CPAP on cardiovascular outcomes in revascularized patients with CAD and concomitant OSA. In the current secondary analysis, we aimed to determine the occurrence of REM-predominant OSA and its association with daytime symptoms, depressive mood, anxiety, and quality of life at baseline in the RICCADSA cohort.

## 2. Materials and Methods

### 2.1. Study Design and Participants

The RICCADSA cohort has been described in detail previously [15,17]. In brief, patients with CAD who underwent percutaneous coronary intervention (PCI) or coronary artery bypass grafting (CABG) in the Skaraborg County of West Götaland, Sweden, were recruited for the main trial between 2005 and 2010, and the follow-up for the primary outcomes was completed in May 2013 [15]. As illustrated in Figure 1, 399 of 511 patients had OSA based on an Apnea-Hypopnea-Index (AHI) ≥15/h on a home sleep apnea testing (HSAT) in the entire RICCADSA cohort. A polysomnography (PSG) was conducted in a hospital in the OSA group before the study started [17]. For the current protocol, only the patients with a REM sleep duration of at least 30 min were included. REM-predominant OSA was defined as a REM-AHI/non-REM-AHI ≥ 2 [3]. None of the patients were on treatment for OSA.

### 2.2. Questionnaires

Epworth sleepiness scale (ESS): The ESS questionnaire [18] was used to determine subjective daytime sleepiness. As previously described, the ESS questionnaire contains eight questions to address the possibility of dozing off under eight different conditions in the past month. A cut-off value of 10 out of 24 was used for categorizing the patients with excessive daytime sleepiness in the RICCADSA trial [17].

Zung Self-rated Depression Scale (SDS): The Zung SDS is a commonly accepted survey that provides both a total score and a categorical rating of depression [19]. In brief, 20 questions are included with a score ranging from 1 to 4 points, which gives a total raw score from 20 to 80. As previously described in detail, the raw score is multiplied by 1.25, and the total result ranges from 25 to 100. The subjects with a score below 50 were categorized as normal, and the patients with a score of 50 or more were classified as having depression [19]. 

Zung Self-rated Anxiety Scale (SAS): The Zung SAS is a Likert scale that compromises 20 items to measure physiological and psychological symptoms [20]. As in the Zung SDS, each item is rated on a 4-point scale from 1 (none, or a little of the time) to 4 (most, or all of the time), and the raw scores (range 20–80) are converted into index scores (range 25–100) by multiplying 1.25 [20]. Individuals with a SAS score of at least 45 points were defined as having clinically significant anxiety [21]. 

Functional Outcomes of the Sleep Questionnaire (FOSQ): FOSQ is a survey regarding sleep-related symptoms, divided into five dimensions, including activity level, vigilance, intimacy, and sexual relationships, general productivity, and social outcome [22]. Item responses range from no difficulty (4 points) to extreme difficulty (1 point). The total score is calculated as the sum of the subscale scores, with a total score ranging from 5 to 20. Lower scores suggest function impairment, and the clinically significant impairment in the FOSQ is defined as a total score of fewer than 17.9 points [23]. 

Short Form-36 Health Survey (SF-36): Data regarding the Health-Related Quality of Life (HRQoL) were collected with the SF-36, which measures eight different domains (Physical Functioning, Role Physical, Bodily Pain, General Health, Vitality, Social Functioning, Role Emotional, Mental Health) and two summary scores (Physical Component Summary [PCS], Mental Component Summary [MCS]) [24]. An SF-36 questionnaire is a subjective tool that reflects the individual’s well-being preceding four weeks prior to the time of the survey. Domain and summary component scores range from 0 to 100; higher scores correspond to better health status or well-being. 

### 2.3. Sleep Recordings

For the HSATs, the Embletta^®^ Portable Digital System device (Embla, Broomfield, CO, USA) was used. As explained previously [15], the HSAT system included a nasal pressure detector, two respiratory inductance plethysmography belts (RIP) for thoracoabdominal movements and body position, and a finger pulse-oximeter for heart rate and oxyhemoglobin saturation (SpO_2_). Apnea was defined as at least ≥90% cessation of airflow and hypopnea was defined as a 50% reduction, at least, in thoracoabdominal movement and nasal pressure amplitude for ≥10 s [25]. Additionally, the total number of significant falls in SpO_2_ (≥4% from the immediately preceding baseline) was scored and the oxygen desaturation index (ODI) was determined as the number of significant desaturations per hour of estimated sleep. Obstructive events with a clear reduction in RIP belts and in the nasal pressure amplitude for at least 10 s were also recorded as hypopneas if there was a significant desaturation [25]. For the overnight PSG in-hospital for the OSA group, before the study started, a computerized recording system (Embla A10^®^, Embla, Broomfield, CO, USA) was used. As previously described [17], the PSG system included sleep monitoring through three-channel electroencephalography (EEG [C4/A1, C3/A2, CZ/A1]), two-channel electrooculography (EOG), one-channel submental electromyography (EMG), bilateral tibial EMG and two-lead electrocardiogram (ECG) in addition to the cardiorespiratory channels as described for the Embletta system above [17]. PSG recordings were scored by an observer blinded to clinical data and baseline screening results from the previous HSAT recordings. Obstructive events on the PSGs were scored according to the same criteria applied to the HSATs. 

### 2.4. Comorbidities

As described previously, baseline anthropometrics, smoking habits, and medical history of the entire study population were obtained from the medical records [15]. Obesity was defined as a body mass index (BMI) ≥ 30 kg/m^2^, and abdominal obesity was defined as the waist-to-hip ratio (WHR) ≥ 0.9 for men and WHR ≥ 0.8 for women, respectively [26].

### 2.5. Statistical Analysis

The Shapiro–Wilk test was used to test the normality assumption of the current data for all variables. Descriptive data are shown as means and standard deviation (SD), or median with interquartile ranges (IQR) for continuous variables, and as a percentage for categorical variables. For comparison between groups, an independent sampled student *t*-test or, when appropriate, the Mann–Whitney U test was used. The Chi-square test was used for the comparison of categorical variables. Pearson’s correlation analysis was used to determine the relationship between REM-AHI levels and the other continuous variables. A logistic regression model was applied to determine variables associated with REM predominant OSA. Age, BMI, sex, percentage of stage 3 of non-REM-sleep (slow-wave sleep [SWS]) and AHI (Model 1) or ODI (Model 2), as well as significant associates (if any) in the univariate analysis, were entered into the multivariate models. Linear univariate regression analysis was used to evaluate the association between REM-AHI and the continuous variables of the ESS, FOSQ, SF-36, Zung SDS, and SAS scores, respectively, and the significant results were entered into the multivariate models with age, BMI, and female sex covariates. The significant variables associated with REM-AHI were tested separately against non-REM-AHI to address if the REM-AHI-related associations were exclusive for the REM sleep stage. All statistical tests were two-sided, odds ratios (ORs) with 95% confidence interval (CI) were reported, and a *p*-value < 0.05 was considered significant. Statistical analysis was performed using the Statistical Package for Social Sciences, version 22.0 for the Windows^®^ system (SPSS^®^ Inc., Chicago, IL, USA).

## 3. Results

A total of 399 participants with OSA were eligible for the current protocol. After excluding patients with REM sleep duration less than 30 min (*n* = 107), total sleep time less than 4 h (*n* = 1), and AHI less than 5 events/h on the PSG (*n* = 5), 286 (mean age 63.7 ± 7.9 years, 86.1% men) remained as the final study population (Figure 1). 

In all, 73 (25.5%) of the study cohort had REM-predominant OSA. As shown in Table 1, there were more women among the participants with REM-predominant OSA compared to those with non-stage-specific OSA (26.0% vs. 9.9%; *p* = 0.001). Average BMI was significantly higher, and obesity was more common (42.5% vs. 25.4; *p* = 0.003) in the REM-predominant group, whereas there were no significant differences between the groups regarding age and comorbidities. 

Table 2 demonstrates that sleep efficiency, as well as the duration and proportion of slow-wave sleep (SWS), was significantly higher in the REM-predominant group, whereas the non-stage specific OSA group had more severe OSA in terms of total AHI and oxygenation indices. Per selection criteria requiring at least 30 min of REM sleep, the REM-dependent OSA group tended to have longer REM sleep time and significantly more severe REM-AHI than the non-stage specific group. 

As shown in Table 3, there were no significant differences between the groups regarding EDS, functional outcomes, mood, and HRQoL measures at baseline. 

In unadjusted logistic regression analyzes, there were significant positive relationships between REM-predominant OSA and female sex, obesity, BMI, and SWS, whereas AHI and ODI were inversely correlated (Table 4). In the adjusted models, female sex, obesity, and BMI, as well as AHI and ODI, remained significant determinants of REM-predominant OSA (Table 4).

As illustrated in Figure 2A, there was significant inverse relationship between REM-AHI and age (r = −0.124, *p* = 0.035), whereas the correlation with BMI was positive (Figure 2B; r = 0.351, *p* < 0.001). 

Other significant correlates of REM-AHI were Zung SDS scores (Figure 3A; r = 0.154, *p* = 0.010) as well as the SF-36 subdomains, Vitality (Figure 3B; r = −0.126, *p* = 0.022) (Figure 3B), Mental Health (Figure 3C, r = −0.148, *p* = 0.013), and MCS (Figure 3D, r = −0.142, *p* = 0.013). No significant associations were found between REM-AHI and other SF-36 domains as well as ESS, FOSQ scores, and Zung SAS scores (data not shown). 

In the adjusted models, the association of REM-AHI with Zung SDS, Vitality, and Mental Health scores disappeared. However, the relationship with MCS remained significant after including age, BMI, and female sex into the model (β-coefficient −2.20, %95 CI −0.56, −0.03; *p* = 0.028) (Table 5). Non-REM-AHI was not associated with the aforementioned variables (r = 0.101, *p* = 0.094 for Zung SDS; r = −0.080, *p* = 0.181 for Vitality; r = −0.106, *p* = 0.077 for Mental Health; and r = −0.101, *p* = 0.098 for MCS). 

## 4. Discussion

The current analysis demonstrated that REM-predominant OSA was significantly associated with female sex and obesity in this revascularized CAD cohort. Despite a lower degree of OSA in terms of AHI and oxygenation indices, the patients with REM-predominant OSA had similar characteristics regarding comorbidities, EDS, functional outcomes, mood, and QoL measures as did the patients with non-stage specific OSA. Further analyzes revealed significant associations of the REM-AHI with depressive mood scores as well as SF-36 subdomains Vitality and Mental Health, which all, however, disappeared in the multivariate models. Only the SF-domain MCS remained significantly associated with REM-AHI independent of age, BMI, and female sex.

To our knowledge, this is the first study to evaluate the association of REM-predominant OSA with excessive sleepiness, functional outcomes, anxiety, and depression, as well as the quality of life measures in a CAD cohort. In prior literature, Zinchuk and colleagues have reported that adults with REM-predominant OSA in sleep clinic cohorts are usually younger women with more efficient sleep and longer total sleep time compared to patients with non-stage specific OSA [3]. Our results suggesting an inverse relationship between REM-AHI and age as well as the dominance of female sex among the REM-predominant OSA supports those findings to be valid also in patients with CAD. Our results are in line with the findings from prior sleep-clinic-based studies, which showed that REM-predominant OSA was not related to excessive sleepiness as measured by multiple sleep latency tests [27,28], as well as with the findings from the large population-based Sleep Heart Health Study, which showed no relationship between REM-AHI and ESS scores after adjustment for confounding factors such as age, sex, and race [11]. 

Few studies addressed the functional outcomes in patients with REM-predominant OSA. Though the definition of hypopneas, as well as REM-predominant OSA, differed, the total FOSQ score tended to be lower among patients with REM-predominant OSA compared to non-stage specific OSA in 2 studies [29,30]. Our results do not support those findings, which might reflect the differences in clinical characteristics of sleep clinics vs. cardiac cohorts, as the patients from sleep clinics are usually symptomatic and have lower FOSQ scores at baseline. The average total FOSQ score was around 12 units in both studies [29,30], whereas our cohort demonstrated values of around 18 units, which suggests that many patients with CAD are with concomitant OSA are asymptomatic and do not have impairment in sleep-related functional outcomes.

Changes in sleep architecture regarding REM sleep have been associated with depressive mood [8,9] and anxiety [10]. In general, sleep pattern changes among patients with depression consist of impaired sleep onset and maintenance, early morning awakening, as well as reduced deep sleep and disinhibition of REM sleep [8]. Sleep disturbances are prevalent in some mental diseases, such as anxiety disorders, and anxiety disorders have been reported to be more common in REM-related OSA in a sleep clinic cohort [10], but we did not observe any difference in the anxiety scores in our CAD cohort. Of note, the Hospital Anxiety-Depression Scale was used in that study [10], whereas Zung SDS and Zung SAS scores were conducted in the current protocol. We found a significant association between REM-AHI and the Zung SDS scores, but this association disappeared after adjustment for the female sex.

Prior studies regarding the association of REM-predominant OSA with the QoL measures have not been consistent. The findings from the Sleep Heart Health Study [11] suggested that NREM-AHI but not REM-AHI was significantly associated with impairment in the PCS and MCS after adjustment for the confounding factors. The inclusion of individuals with a REM sleep time of less than 30 min, as well as the statistical analysis including both REM-AHI and NREM-AHI in the same model (collinearity), might explain some of the differences between our findings and the results of that population-based study [11]. We found significant linear correlations between REM-AHI and Mental Health and Vitality scores, but these associations were not significant anymore in the multivariate analysis. Notwithstanding, the association of the REM-AHI with the MCS remained significant after adjustment for age, BMI, and female sex. Given the absence of a significant association between non-REM-AHI and MCS, our findings suggest that the decline in MCS is exclusive to the REM sleep stage. 

It should be noted that there have been several definitions of REM-dependent OSA in literature: (I) overall AHI ≥ 5 and REM-AHI/NREM-AHI ratio ≥ 2; (II) overall AHI ≥ 5, REM-AHI/NREM-AHI ratio ≥ 2, and NREM-AHI < 15; or (III) overall AHI ≥ 5, REM-AHI/NREM-AHI ratio ≥ 2, NREM-AHI < 15, and at least 10.5 min of REM sleep duration [31]. There have also been studies including only subjects with at least 30 min of recorded REM sleep [32] to reduce the possibility of exaggerating the effect of REM-predominant OSA in individuals for short REM duration, which is also applied in the current study. However, the strict criterion brings a potential bias, as the excluded patients are probably the ones with severe OSA (overall AHI ≥ 30), who otherwise would belong to the non-stage-specific OSA. 

Notwithstanding, the significant association of REM-AHI with the MCS despite the lower degree of OSA severity highlights the need for a more complete characterization of each patient’s pathophysiology and a sex-stratified approach in the management of patients with CAD with concomitant OSA. This may be particularly important for increasing adherence to CPAP treatment to cover the early morning hours before awakening, thus treating the whole REM sleep period for better MCS outcomes. Given that there is a decrease in upper respiratory muscle tone in the REM period [33], which results in lower respiratory drive, and that longer periods of respiratory events cause more severe hypoxemia [34], a dose-response relationship between REM-AHI and hypertension has also been suggested [35]. However, more research with larger studies considering individual risk factor profiling in refining treatment strategies in OSA phenotypes is still urged. 

We should acknowledge certain limitations. Firstly, the power estimate for the entire RICCADSA cohort was conducted for the primary outcome and not for the secondary outcomes assessed in this subprotocol. Secondly, the EDS definition was based on the ESS threshold, which may not be precise in a CAD population. Thirdly, our results are not generalizable to adults with OSA in the general population or in sleep clinic cohorts. 

## 5. Conclusions

To conclude, female sex and BMI were related to REM-predominant OSA in this revascularized cohort. MCS was inversely associated with REM-AHI in the multivariate model. Further research is needed to establish better whether these patients will obtain any benefit from long-term CPAP therapy and whether sex-stratified treatment strategies should be applied for patients with CAD and REM-predominant OSA. 

## Figures and Tables

**Figure 1 jcm-11-04402-f001:**
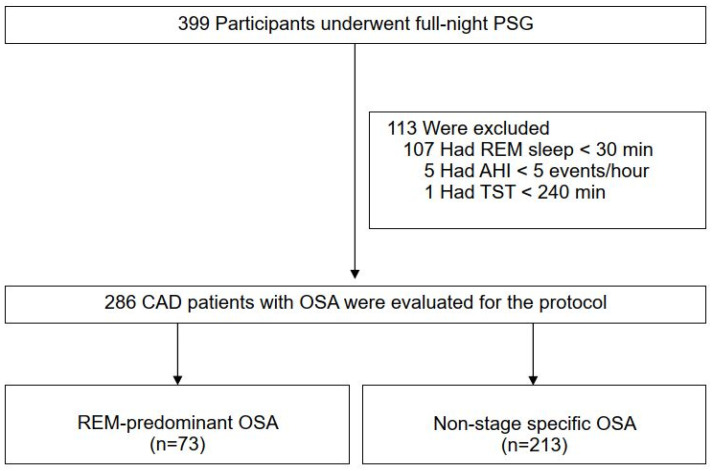
Consort flow chart of the study sample. *Definition of abbreviations*: AHI = apnea-hypopnea index; CAD = coronary artery disease; OSA = obstructive sleep apnea; PSG = polysomnography; REM =rapid eye movements; TST = total sleep time.

**Figure 2 jcm-11-04402-f002:**
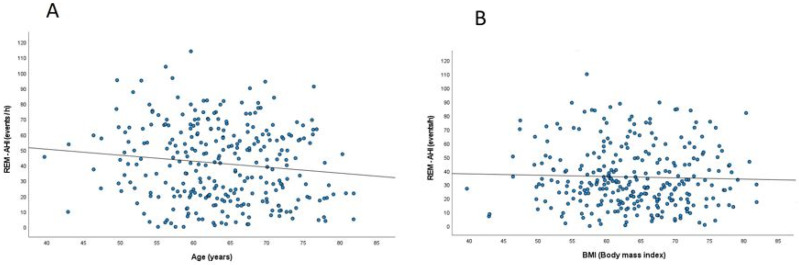
(**A**). Pearson correlation between REM-AHI levels and age, (**B**). Pearson correlation analysis between REM-AHI and BMI. *Definition of abbreviations*: AHI = apnea-hypopnea-index; BMI = body mass index; REM = rapid eye movements.

**Figure 3 jcm-11-04402-f003:**
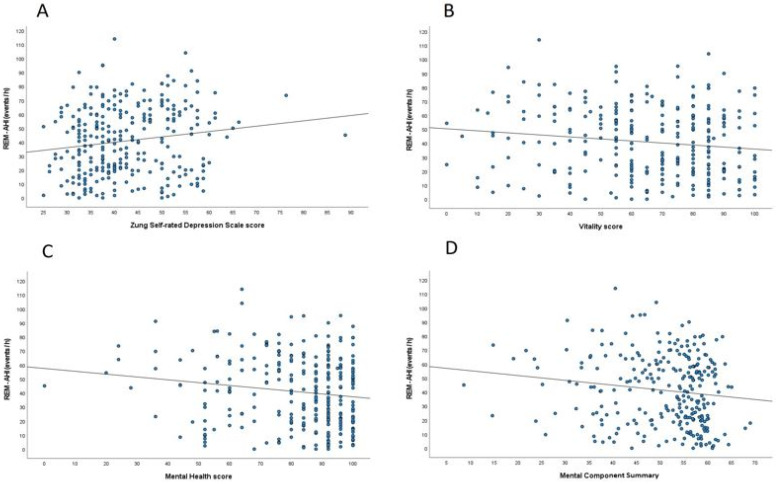
(**A**). Pearson analysis between REM-AHI and Zung Self-rated Depression Scale score. (**B**). Pearson analysis between REM-AHI and Vitality Score. (**C**). Pearson analysis between REM-AHI and Mental Health score. (**D**). Pearson analysis between REM-AHI and Mental Component Summary score. *Definition of abbreviations*: AHI = apnea-hypopnea-index; REM = rapid eye movements.

**Table 1 jcm-11-04402-t001:** Baseline demographic and clinical characteristics of the study population (*n* = 286).

	REM-Predominant OSA N = 73	Non-Stage Specific OSA N = 213	*p*-Value
Age, years	63.2 ± 7.5	63.8 ± 8.1	0.539
Age ≥ 65 years, %	43.8	43.2	0.924
Female sex, %	26.0	9.9	0.001
BMI, kg/m^2^	29.0 (26.3–32.2)	28.1 (25.8–29.9)	0.020
Obesity, %	42.5	24.4	0.003
Current smoker, %	16.4	16.4	1.000
CABG at baseline, %	20.5	23.9	0.552
AMI at baseline, %	43.8	51.2	0.279
Former revascularization, %	24.3	21.0	0.554
Hypertension,%	64.4	60.6	0.563
History of atrial fibrillation, %	11.0	16.9	0.225
Diabetes mellitus, %	24.7	22.1	0.648
History of stroke,%	8.2	7.6	0.861
Pulmonary disease,%	5.5	6.1	0.846
Anti-depressive medication, %	5.6	3.4	0.419

*Definitions of abbreviations*: AMI = acute myocardial infarction; BMI = body mass index; CABG = coronary artery by-pass grafting; OSA = obstructive sleep apnea; REM = rapid eye movements.

**Table 2 jcm-11-04402-t002:** Polysomnographic findings of the study population (*n* = 286).

	REM-Predominant OSA N = 73	Non-Stage Specific OSA N = 213	*p*-Value
TST, min	430.5 (374.8–469.3)	413.3 (371.7–451.5)	0.320
Sleep Onset, min	10.0 (5.3–14.1)	9.3 (5.7–16.6)	0.702
Sleep Efficiency, % of TST	84.6 (78.3–91.0)	82.4 (75.5–88.3)	0.041
SWS, min	32.5 (3.5–59.5)	22.0 (0.5–47.3)	0.039
SWS, % of TST	7.8 (0.9–15.3)	5.6 (0.1–11.5)	0.044
REM sleep, min	63.5 (47.5–79.8)	59.5 (42.0–71.8)	0.077
REM sleep, % of TST	15.7 (12.3–19.4)	14.4 (10.9–18.0)	0.151
AHI, events/hour	22.6 (13.9–31.9)	36.6 (24.4–55.7)	<0.001
REM—AHI, events/hour	53.5 (33.4–69.0)	34.5 (18.2–56.8)	<0.001
Non-REM—AHI, events/hour	16.1 (9.7–27.4)	38.1 (25.9–57.1)	<0.001
ODI, events/hour	10.2 (6.7–19.9)	19.2 (9.2–31.5)	<0.001
Average SpO_2_ %	94.0 (93.2–95.1)	93.9 (92.8–94.8)	0.098
Nadir SpO_2_ %	82.0 (78.0–86.5)	84.0 (79.0–87.0)	0.289
SpO_2_ < 90%, min	5.1 (1.2–16.9)	5.1 (1.0–24.0)	0.473
SpO_2_ < 90%, % of TST	1.5 (0.3–4.8)	1.2 (0.2–5.3)	0.820
Heart rate, beats/min	58.8 (53.4–64.8)	57.0 (51.3–63.2)	0.167

*Definitions of abbreviations*: AHI = apnea-hypopnea-index; ODI = oxygen desaturation index; OSA = obstructive sleep apnea; REM = rapid eye movements; SpO_2_ = oxyhemoglobin saturation; SWS = slow wave sleep (stage 3 of non-REM sleep); TST = total sleep time.

**Table 3 jcm-11-04402-t003:** Comparison of excessive sleepiness, functional outcomes, depression, anxiety, and quality of life measures of the study participants (*n* = 286) polysomnographic findings of the study population (*n* = 286).

	REM-Predominant OSA N = 73	Non-Stage Specific OSA N = 213	*p*-Value
ESS score	8 (4.0–11.0)	8.0 (5.0–11.0)	0.504
EDS (ESS score ≥ 10), %	42.5	39.9	0.598
FOSQ scores			
General Productivity	3.8 (3.6–4.0)	4.0 (3.7–4.0)	0.216
Social Outcome	4.0 (4.0–4.0)	4.0 (4.0–4.0)	0.580
Activity level	3.8 (3.2–3.9)	3.7(3.3–3.9)	0.872
Vigilance	3.7 (3.4–4.0)	3.7 (3.3–4.0)	0.891
Intimate relationship	3.5 (3.0–4.0)	3.8 (3.0–4.0)	0.988
Total score	18.5 (17.6–19.4)	18.7 (17.3–19.6)	0.594
Zung SDS score	38.8 (33.4–50.0)	41.3 (35.0–50.0)	0.244
Depression (Zung SDS score ≥ 50), %	31.4	28.3	0.618
Zung SAS score	36.3 (31.3–43.8)	38.8 (33.8–41.3)	0.211
Anxiety (Zung SAS score ≥ 45), %	21.4	15.9	0.294
SF-36 domains			
Physical Functioning	85.0 (65.0–95.0)	83.3 (68.3–90.0)	0.577
Role Physical	75.0 (25.0–100.0)	75.0 (25.0–100.0)	0.575
Bodily Pain	84.0 (61.0–100.0)	74.0 (51.0–100.0)	0.090
General Health	70.0 (57.0–82.0)	67.0 (52.0–82.0)	0.610
Vitality	70.0 (55.0–85.0)	70.0 (50.0–85.0)	0.782
Social Functioning	100.0 (87.5–100.0)	100.0 (75.0–100.0)	0.230
Role Emotional	100.0 (66.7–100.0)	100.0 (66.7–100.0)	0.307
Mental Health	88.0 (75.0–96.0)	88.0 (72.0–96.0)	0.379
PCS	46.2 (39.4–52.9)	46.0 (37.6–52.6)	0.690
MCS	55.1 (46.9–58.7)	54.2 (45.3–57.9)	0.335

*Definitions of abbreviations*: ESS = Epworth sleepiness scale; FOSQ = functional outcomes of sleep questionnaire; MCS = mental component summary; OSA = obstructive sleep apnea; PCS = physical component summary; REM = rapid eye movements; SAS = self-rating anxiety scale; SDS = self-rating depression scale; SF = short form.

**Table 4 jcm-11-04402-t004:** Unadjusted and adjusted ORs (95% CIs) for variables associated with REM predominant OSA.

Variables	OR	95 CI%	*p*-Value
**Unadjusted**			
Age, years	0.98	0.96–1.02	0.538
Female sex	3.22	1.61–6.41	0.001
BMI	1.09	1.02–1.16	0.011
Obesity	2.29	1.30–3.99	0.004
ESS	0.96	0.90–1.03	0.316
EDS (ESS ≥ 10)	1.16	0.67–1.98	0.598
SWS, % of TST	1.04	1.01–1.08	0.015
AHI, events/h	0.95	0.93–0.97	<0.001
ODI, events/h	0.96	0.94–0.98	<0.001
Nadir SpO_2_ %	0.98	0.95–1.02	0.346
SpO_2_ < 90%, min	0.99	0.98–1.00	0.098
SpO_2_ < 90%, % of TST	0.97	0.93–1.01	0.115
Current smoker	1.00	0.48–2.05	0.999
Baseline AMI	0.75	0.44–1.27	0.280
Hypertension	1.18	0.68–2.05	0.563
History of atrial fibrillation	0.61	0.27–1.37	0.228
Diabetes mellitus	1.16	0.62–2.16	0.649
Stroke	1.09	0.41–2.90	0.861
Pulmonary disease	0.89	0.28–2.83	0.846
Zung SDS score	0.99	0.96–1.01	0.297
Zung SAS score	0.98	0.94–1.01	0.217
PCS	1.01	0.98–1.04	0.507
MCS	1.01	0.99–1.04	0.355
**Adjusted**			
**Model 1**			
Age, years	0.98	0.94–1.02	0.374
Female sex vs. Male sex	4.64	1.85–11.64	0.001
BMI, kg/m^2^	1.17	1.07–1.28	<0.001
SWS, % of TST	0.99	0.96–1.03	0.716
AHI, events/h	0.93	0.91–0.95	<0.001
**Model 2**			
Age, years	0.98	0.94–1.02	0.359
Female sex vs. Male sex	2.96	1.30–6.72	0.009
BMI, kg/m^2^	1.14	1.05–1.23	0.003
SWS, % of TST	1.01	0.98–1.05	0.606
ODI, events/h	0.95	0.92–0.97	<0.001

*Definitions of abbreviations:* AMI = acute myocardial infarction; AHI = apnea-hypopnea-index; BMI = body mass index; CI = confident interval; ESS = Epworth sleepiness scale; MCS = mental component summary; ODI = oxygen desaturation index; OSA = obstructive sleep apnea; PCS = physical component summary; REM = rapid eye movements; SAS = self-rating anxiety scale; SDS = self-rating depression scale; SF = short form; SWS = slow wave sleep (stage 3 of non-REM sleep).

**Table 5 jcm-11-04402-t005:** Variables associated with REM-AHI in adjusted linear regression analyzes.

	β Coefficient	95% CI	*p*-Value
**Model 1**			
Age	−0.04	−0.46, −0.25	0.544
BMI	0.30	1.15, 2.61	<0.001
Zung SDS score	0.12	0.01, 0.57	0.045
**Model 2**			
Age	−0.08	−0.59, 0.11	0.177
BMI	0.27	0.95, 2.38	<0.001
Zung SDS score	0.09	−0.05, 0.50	0.115
Female sex	0.23	7.95, 23.70	<0.001
**Model 3**			
Age	−0.08	−0.59, 0.12	0.185
BMI	0.27	0.99–2.42	<0.001
Vitality	−0.07	−0.19, 0.05	<0.001
Female sex	0.24	9.30, 25.01	0.231
**Model 4**			
Age	−0.08	−0.59, 0.12	0.192
BMI	0.27	1.00, 2.42	<0.001
Mental Health	−0.09	−0.28, 0.02	0.094
Female sex	0.24	8.99, 24.64	<0.001
**Model 5**			
Age	−0.08	−0.61, 0.11	0.168
BMI	0.28	1.06, 2.49	<0.001
MCS	−0.12	−0.55, −0.02	0.033
Female sex	0.25	9.81, 25.82	<0.001

*Definition of abbreviations*: AHI = apnea-hypopnea-index; BMI = body mass index; CI = confident interval; MCS = mental component summary; OSA = obstructive sleep apnea; REM = rapid eye movements; SDS = self-rating depression scale.

## Data Availability

Individual participant data that underlie the results reported in this article can be obtained by contacting the principal investigator of the RICCADSA trial; yuksel.peker@lungall.gu.se.

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
