# Peer review of "REM-Predominant Obstructive Sleep Apnea in Patients with Coronary Artery Disease"

_jcm, 2022, doi:10.3390/jcm11154402_

Round 1

Reviewer 1 Report

This study aimed to investigate gender differences in depression and quality of life in patients with CAD and REM-predominant OSA. There are several major flaws :

1. The rationale is not clear. This study does not clearly distinguish between REM-predominant OSA and REM-AHI. The narrative of the study sample size is also very unclear. The statistical method is also unreasonable, and it does not check whether the data conform to the normal distribution.

2. This study does not indicate whether the included analysts received CPAP therapy or other possible OSA therapy.

3. The authors did not perform non-REM-AHI data analysis. Therefore, the analysis results for REM-AHI may be unreasonable, and the conclusions may also be incorrect.

4. This study hopes to explore gender differences, but in the analysis, gender is not stratified and discussed, so it is impossible to know the status of gender differences.

5. From the analysis data, how do you know the relationship between REM-AHI and depression, vitality, and mental health dependent on female sex?

Reviewer 2 Report

Dear authors, 

thank you very much for the opportunity to review this well written article about REM predominant  OSAin CAD patients and the specific sex differences and quality of life. 

Overall, the article is well written and clear about the research question. It is well notable that it discusses a secondary outcome from a previously executed study, which makes the conclusion of the study somewhat vague. 

It is questionable whether this article contributes to the daily practice of the different physicians working in the field of sleep diseases or cardiology. But in its niche, this article is well written and explores the data sufficiently to finalize with clear conclusions. 

Perhaps the article would be more clearly if certain questionnaires were deleted, e.g. SDS and SAS. It would highlight the more important variables and could give a better overview of the study population, with only the ESS, (maybe FOSQ), sleep parameters, and SF-36. 

Abstract: sometimes difficult to read, because of the many variables illustrated. 

Introduction: Perhaps include something about the treatment for CAD. 

M&M: IN this section a good insight is given about the population, that is lacking in the abstract: already treated patient for CAD with PCI or CABG (was this successful and how did the CAD parameters improved on this. Moreover, if this was successfully executed what implications did this had for the outcome of this study?)

In the SF-36 section there is an extra right parenthesis. 

Results: outcome of the CAD treatment are lacking, so it is not clear if this has an effect on the OSA parameters also. 

Figure 1 is displaying a lot of abbreviations that are not used in the figure and are confusing in this context. 

Discussion: The results are extensively assessed on the present literature and discussed. But the limitation in this study is selection bias in the CAD groups and the different outcomes for these patients. This is not presented and could play a major role in the different outcomes in the OSA parameters, because of overlapping symptoms, especially in the HRQoL aspects, the ESS and the FOSQ. Please add more detailed information about this.

Wish the authors all the best with finalizing this manuscript. 

Kind regards. 
